# Readiness of the Polish Crisis Management System to Respond to Long-Term, Large-Scale Power Shortages and Failures (Blackouts)

**Dariusz Majchrzak [1], Krzysztof Michalski [2,*] and Jacek Reginia-Zacharski [3]**

1 Security and Defence Faculty, War Studies University, 00910 Warsaw, Poland; d.majchrzak@akademia.mil.pl
2 Faculty of Management, Rzeszów University of Technology, 35959 Rzeszów, Poland
3 Faculty of International & Political Studies, University of Lodz, 90127 Łódź, Poland; jacek.reginia@uni.lodz.pl
* Correspondence: michals@prz.edu.pl; Tel.: +48-537-111-727

**Abstract:** Large-scale failures of electric power systems (blackouts) have been the subject of intensive research in most countries for several years. This research aims primarily at seeking solutions to improve the reliability of the operation of power systems and the development of effective strategies to protect critical infrastructure from the effects of energy shortages and power cuts. In contrast, systematic research on crisis management and civil protection under conditions of prolonged blackout has been undertaken in Europe only recently, and these extremely important aspects of energy security have been delayed by the COVID-19 crisis. The ability of the Polish crisis management system to cope with the consequences of long-term, large-scale shortages and interruptions in the supply of electricity, as well as the consequences of possible failures in this field, has not been systematically examined to date. This issue is of growing strategic importance, not only from the point of view of security and defence policy, but also economic cooperation in Central and Eastern Europe. Poland's infrastructural security must be considered in a broad regional and supra-regional context. A long-term lack of electricity in a large area of Poland would undermine the stability of the entire national security system, destabilising the region and supranational security systems. Apart from objective reasons, intentional attacks on the links of such a chain cannot be ruled out. Poland is the leader of this region, a frontline country in the NATO-Russia conflict, as well as a liaison state that provides the Baltic states—being EU and NATO members—with a land connection to Western Europe. In view of the growing risk of blackout, the importance of the problem and the existence of a cognitive gap in this field, we evaluated the Polish crisis management system in terms of its ability to respond to the effects of a sudden, long-term, large-scale blackout. Methodologically, we adopted a systems approach to security management. In order to estimate the consequences of a blackout, we used analogue forecasting tools and scenario analysis. By analysing previous crisis situations caused by blackouts and local conditions of vulnerability to such events, we formulated basic preparedness requirements that a modern crisis management system should meet in the face of the growing risk of blackouts. A review of strategic documents and crisis planning processes in public administration allowed us to identify deficits and weaknesses in the Polish crisis management system. On this basis we formulated recommendations whose implementation shall improve the ability of the national security system to face such challenges in the future.

**Keywords:** energy safety and security; homeland security; crisis-management; critical infrastructures; civil defence/protection; disaster resilience

## 1. Introduction

Under the influence of a complex set of factors, the interconnected electricity systems in Europe and the component systems of most countries are now balancing on the threshold of failure, making the spectre of widespread, prolonged blackouts, which threaten humanitarian catastrophes in modern networked societies that are hopelessly dependent on IT and

electricity, increasingly real. The high burdens currently imposed in preventing the collapse of interconnected power systems and the number of interventions necessary to maintain their stability mean that these systems are in a permanent state of stress, increasing the risk of catastrophic outages that can trigger unpredictable, spatio-temporal chains of failures that are difficult to control with the technical means available. In most countries, the cost of managing operational bottlenecks and the cost of handling dangerous incidents in national power systems has increased several times over the last ten years. Currently, in Austria alone, almost one million Euros are spent every day on ensuring the security and stability of the electricity system, which is almost twice the annual expenditure incurred for this purpose in 2011. These are the direct costs incurred just to ensure system security and prevent system collapse. These sums do not include the costs of infrastructure investment, but merely the costs of operating mechanisms for safely disconnecting power stations from the grid and reconnecting them. In 2018, interventions necessary to maintain the stability of the national power system in the face of dangerous incidents were undertaken on more than 300 days in Austria [1]. Due to the structural similarities of the electricity supply systems (including the lack of nuclear energy), Austria must serve as a reference point here, in the absence of similar data relating to the situation in Poland.

Under the influence of global trends linked to energy transformation (Energiewende) and specific national circumstances, the probability of catastrophic blackouts, i.e., a total blackout of the power system over a large area, is increasing dangerously in Poland. The deep structural changes in Poland's electricity system that are taking place now and are expected in the near future, necessitated by the European Green Deal policy, assume first and foremost, the gradual replacement of stable, conventional sources of energy which possess large reserves of power with distributed renewable power sources that pulse and provide energy in a manner that is not connected to the power grid. Their operation and the preservation of network stability requires the intelligent development and transformation of power infrastructures into complex, autonomous cyber-physical systems, which, in turn, will cause a sharp increase in their structural complexity, fostering unpredictable behaviour [2,3] and making them an easy target for cyber-attacks [4–6]. A prolonged, large-scale blackout is, for any highly developed society hopelessly dependent on electricity for even the simplest of life's activities, undoubtedly one of the most catastrophic cumulative hazard scenarios that still fails to reach the consciousness of most individuals—and this includes those responsible for security, civil protection and emergency management [7]. Just as two years ago, no one seriously considered the possibility of a catastrophic pandemic such as SARS-CoV-2, and no one expected how suddenly it would be able to paralyse the lives of highly developed societies, bringing vital circulatory systems to a standstill for months [8], with the same incredulity political decision-makers and most of public opinion greet expert warnings of the growing risk of large-scale blackouts and apocalyptic visions of the humanitarian catastrophes that the possible occurrence of a particularly long-lasting failure of international scope would cause in societies completely unprepared for networked, cumulative, multi-level emergencies.

The fundamental characteristics of crisis situations caused by a catastrophic blackout are, on the one hand, their large geographical extent and uniformity of impact, and, on the other hand, the limited predictability of their duration, which has a crucial bearing on the accuracy of decisions taken in the first phase of a blackout, on which the success of all subsequent emergency actions largely depends. Unlike focused or isolated emergencies that are caused by natural or industrial disasters, where emergency response involves the appropriate deployment of resources at the disposal of civil protection services to the places where they are most needed, blackout emergencies affect populations who are residing over a wide area in equal measure, and each local government unit within the scope of the emergency faces similar problems of its own, usually using inadequate resources. Unable to predict in advance how long the crisis situation will last, no one is prepared to share any spare capacity. Due to the large spatial extent, complexity and high severity of the crises, EU mutual assistance mechanisms may be of limited use. If a

large part of Europe is affected by a blackout, there will be many who are eager to activate the EU civil protection mechanism but few who are prepared to provide assistance. The distinctiveness of large-scale power failures as a source of emergencies is that in modern societies which are increasingly dependent on IT, and therefore electricity, uncontrolled cascades of system failures based on technological components—systems critical to the biological survival of the population, security and defence, social development and the economy—can be triggered. Consequently, this can cause spatio-temporally unlimited chains of destructive events which are capable of overcoming every protective barrier invented by man and threatening everything within their range of impact. Therefore, in the context of energy security, which in most countries has recently risen to the status of one of the main priorities of security and defence policy [9], building preparedness and capacity to cope with crisis situations resulting from long-term, large-scale power failures, in addition to preventive measures aimed at the early identification of future needs, planning, self-sufficiency, diversification, security buffers and strengthening resilience, is now of great political importance.

On the eve of the SARS-CoV-2 pandemic, an international consultation on the risk assessment of long-term, large-scale blackouts/blackouts in Europe and how to prepare for the resulting possible disaster and emergency scenarios was held on 7–8 January 2020 at the Austrian Ministry of the Interior hosted by Department II/13 SKKM (Crisis and Disaster Management and Civil Defence Coordination) with the participation of the German and Swiss civil defence and civil protection authorities. Participants in the consultation agreed that existing strategies for dealing with large-scale 'network' crises—strategies inspired in most countries by their experiences with natural disasters and catastrophes—are inadequate and that effective protection against such increasingly likely hazard scenarios requires a reappraisal and reconfiguration of existing crisis management and civil protection systems. New concepts, methodologies, education and information measures are required, as well as new forms of involvement and cooperation. The results of the Austrian consultation prompted us to undertake research into the challenges that the growing risk of catastrophic blackouts poses for crisis management and civil protection under Polish conditions.

This paper presents the preliminary results of the heuristic-structural phase of an extensive, comprehensive study of the possible causes and consequences of catastrophic, persistent, large-scale blackouts, as well as effective ways to maintain continuity of operation of key state functions under blackout conditions and ways to prevent network emergencies from turning into national disasters. We are committed to diagnosing the capacity of the Polish crisis management system to cope with such cumulative crisis situations, which can only be paralleled in terms of the number of casualties by attacks using nuclear weapons. The diagnosis aims to identify errors, gaps and weaknesses in the crisis management system and to develop solutions that are better suited to the requirements of the new types of threats which are capable of triggering cascades of failures leading to cumulative network crises. Based on an analysis of our own experience to date with power failures of a small spatio-temporal extent, the experience of other countries with widespread, long-lasting blackouts, and analogies with cumulative emergencies resulting from knock-on failures initiated by other kinds of factors, such as the ongoing SARS-CoV-2 pandemic, we explain the peculiarity, structural complexity and escalatory dynamics of such emergencies, in the face of which the traditional crisis management and disaster protection systems of most countries capitulate.

In view of the growing risk of bottlenecks in the electricity supply system and widespread and prolonged blackouts, it is necessary to assess the level of preparedness of the Polish crisis management system for hypothetical scenarios of cumulative supply crises, national disasters and secondary threats, before actual crises put the system to a real test. The gaps and weaknesses of the system under consideration identified during the study were used to formulate optimisation recommendations, the implementation of which will allow the crisis management system to be properly calibrated and configured from the point of view of the requirements and challenges posed to Poland's national security

by the far-reaching transformation of the electricity sector. The issues discussed in the article may be of interest, not only to theoreticians and practitioners of crisis management in Poland, but also to researchers dealing with international security—including military security. The security of the entire eastern flank of NATO and a large part of the EU depends to a large extent on the state of Poland's security and the stability of the structures that guarantee it [9]. Poland is not only the leader of the Euroregion, an important buffer in the still tense NATO-Russia relations and the link providing the three Baltic States with a land connection to the EU, but also an increasingly important link in European energy supply chains. In view of Poland's growing international importance—both politically, militarily and economically—the dysfunctions of the Polish crisis management system, which threaten to transform a long-term, large-scale blackout into a national disaster—the effects of which would be felt by many countries linked to Poland by complex networks of mutual dependencies—are becoming important for European security also beyond the primary area of energy security.

## 2. Description of Research Objectives, Sources and Methods Used

Although large-scale power system failures are more and more frequently included in the most important threat scenarios of the near future, Poland, due to local conditions, is particularly exposed to shortages and interruptions in the supply of electricity. The consequences of a blackout and the ability of the national crisis management system to respond to such a situation have not yet been comprehensively studied in Poland. No large-scale, systematic research has been conducted in Poland to date on the readiness of the Polish crisis management system to cope with the effects of a long-term wide area blackout, but this issue has been discussed for several years within the framework of a series of conferences on the technical and non-technical aspects of blackout, organised by the Institute of Electrical Power Engineering of the Poznań University of Technology since 2004 [10] (p. 680), as well as on the margins of many other conferences on energy security, which, due to local factors mainly related to the 'cold gas war', enjoy an unabated boom in Poland. The issue is rarely addressed at a general level in small overviews [10–12], or in the margins of research on the technical aspects of protecting key infrastructures from the consequences of blackouts [13–15], and in the margins of case studies that include explanations of the causes and consequences of specific power failures [16].

In view of the lack of national experience with long-lasting, large-scale blackouts—both real and simulated—and the lack of domestic research on the challenges that cascades of spikes caused by catastrophic blackouts may pose to Polish crisis management services, in forecasting crisis scenarios for Poland and assessing the readiness of the Polish crisis management system to cope with such situations, we used reference research conducted several years ago in Germany [17–21], case studies describing experiences of other countries with catastrophic blackouts [22–33], documentation of national experiences with local blackouts [34], and qualitative analyses of strategic documents containing assumptions of the Polish security policy [35] and analyses of current crisis management plans—both the national plan [36,37] and plans of selected local government units [38–40]. Within the framework of these German projects, the possible causes of blackouts and the vulnerability of modern industrialised and computerised societies to a long-term, large-scale blackout were analysed in detail in terms of the potential for a nationwide catastrophe. In assessing the impact of such events, particular attention has been paid to the impact on specific critical infrastructure sectors, such as telecommunications and IT systems, transport, water supply and sewage disposal, food supply, the health system, the financial system, and state security authorities and public administration [20].

The results of the analysis of the potential effects of a widespread, long-term blackout and the resulting requirements for maintaining business continuity and civil protection were compared with the assessment of the operational capabilities of the Polish crisis management system undertaken on the basis of an analysis of planning documents. As a result, a pessimistic diagnosis was formed regarding the readiness of the Polish crisis

management system to meet the challenges posed by various blackout scenarios, and a number of optimisation recommendations were developed for preparing and responding to this type of crisis situation. Implementing these recommendations will minimise the risk of emergencies caused by a prolonged, large-scale blackout turning into nationwide disasters. The need to change the existing reductionist model of threat perception based on elementarisation, linear-deterministic understanding of causality and belief in calculability of risk, and to replace it with an 'organic', 'network' model, which enables more adequate recognition of complexity and the escalation potential of interconnected crisis situations, as well as more accurate planning, has been recognised as a key action from the point of view of the possibility of managing such crisis situations (1), a gradual move away from the hitherto centralised, sectoralist, bureaucratic, exclusive model of security management, based on rigid hierarchical structures, task parcelling, proceduralisation, routine and a passive, retrospective understanding of responsibility, towards an 'organic', 'network', inclusive management, based on initiative and creativity, popular commitment, intersectoral cooperation and synergies and an active, prospective understanding of responsibility (2), urgently organizing an efficient communication, alert and crisis communication system resistant to prolonged outages (3), ensuring an efficient and robust emergency power supply system with a capacity adapted to the needs of field administration units with the option of fast mobility (4), building an adequate logistical base to ensure the continuity of supply of emergency power supply systems with fuel (5), building barriers and island networks to prevent the spread of cascades of disturbances (6), optimising resource management and improving the capacity of the crisis management system to rapidly and smoothly transition to multivariant emergency modes (7), public awareness campaigns to increase people's self-sufficiency, promote strategic stockpiling for long-term supply crises, develop the ability to cope with such situations without external assistance, and foster an attitude of civic engagement and mutual aid (8).

Similar research initiatives have recently been undertaken in neighbouring countries [41], which proves the topicality of the issues presented below.

## 3. Results

The national crisis management system (CMS) has been operating in Poland in the present format since 2007 under the Act of 26 April 2007 on crisis management [42]. Crisis management is all activities of a public authority which are an element of managing national security, consisting of the prevention of a crisis situation, preparation to take control of such situations by planned actions, responses in the case of crisis situations and the reconstruction of resources and critical infrastructures. Effective crisis management is integration of emergency plans at all levels of government and non-government involvement. Activities at each level (individual, group, community) affect the other levels. It is common to place the responsibility for governmental emergency management with the institutions for civil defence or within the conventional structure of the emergency services. Crisis management consists of four phases: preparation, prevention, response and recovery.

The structure of the crisis management system is related to the administrative division and results from the regulations concerning this area (Figure 1). The overall concept of crisis management in Poland is based on primacy of the territorial arrangement. The System has been developed in Poland in order to deal with different types of crisis situations.

The Crisis Management System is a system of three ordered items that meet the following roles:

- Decision-making body: the Council of Ministers, minister of administration, voivode, leader of the district, mayor, president of the city—have the sovereign decisions assigned to it by law or regulation;
- Advisory body: the Government Team for Crisis Management, voivodship, district and municipal Teams for Crisis Management—identifying strategies in a crisis situation, recommending possible solutions, providing communication between the member's decision-making and planning and coordination;

-   Planning and coordinating body: analyse and evaluate information, prepare options for action and proposals for solutions variants and is responsible for the implementation of the decisions taken, the division of tasks and coordination.

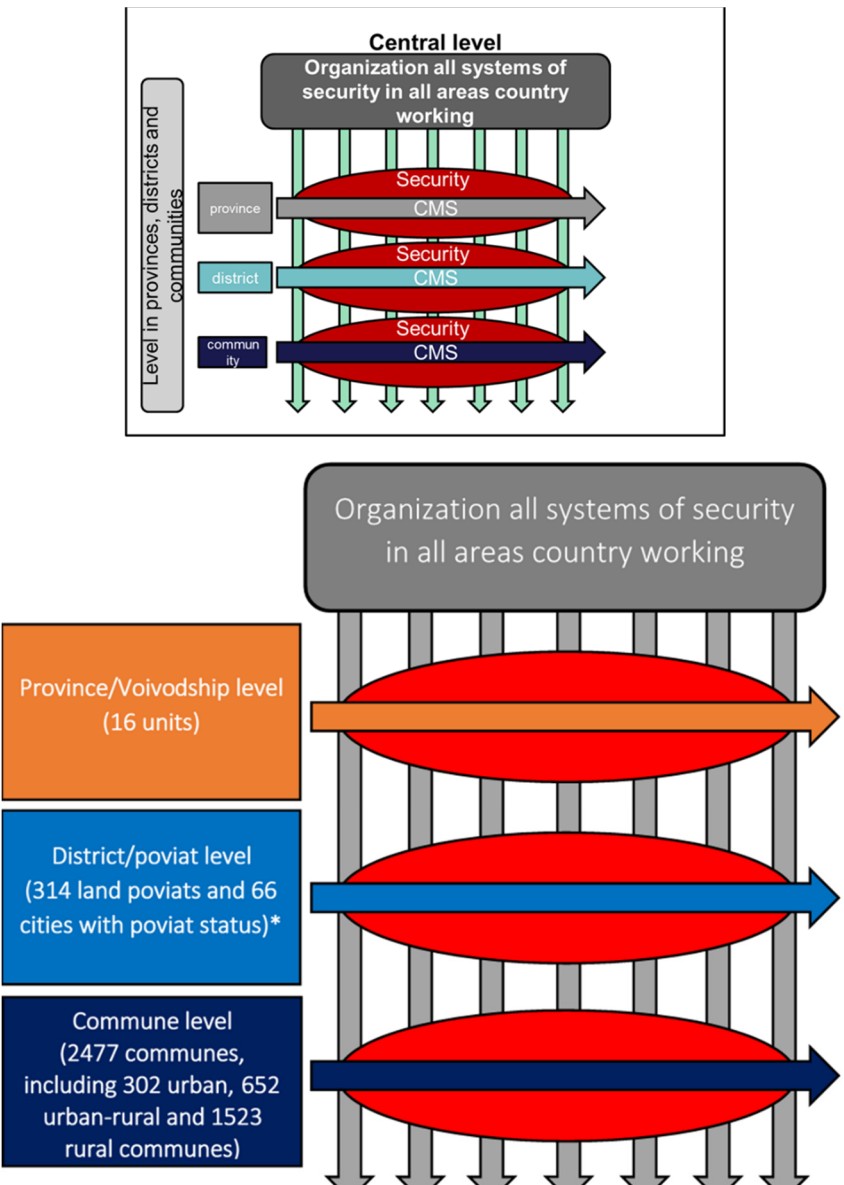

**Figure 1.** The general model of crisis management system (CMS) in Poland (* 1 January 2021).

The Act on Crisis Management specifies competent authorities in matters of crisis management and their objectives and principles of action in this regard. The crisis management system is multi-levelled—the Act introduces four main levels of crisis management: national, voivodship, district and commune. It is also possible to distinguish the ministry level (Table 1).

In the crisis management system, the Management Authority whose task is making decisions, distributing tasks to individual contractors and coordinating activities during an emergency situation, fulfils the leading role. The Management Authority defines the scope of projects, and is also responsible for actions taken during the phases of crisis management, which includes the prevention of a crisis situation, preparation to take effective action, the proper response and return to normal functioning (Figure 2). The subsystem performs its tasks through management, which is the decision making process consisting of capturing, collecting, processing and distributing information.

**Table 1.** The structure of crisis management in Poland.

| Administrative Level | Crisis Management Authority | Opinion Forming and Advisory Body | Crisis Management Office |
|---|---|---|---|
| National | Council of Ministers, Prime Minister | Government Crisis Management Team (RZZK) | Government Center for Security (RCB) |
| Ministerial | Minister in charge of government administration department | Ministerial Crisis Management Team | Ministerial Crisis Management Center |
| Province/Voivodship | Voivod | Voivodship Crisis Management Team (WZZK) | Voivodship Crisis Management Center (WCZK) |
| District | District (Foreman) | District Crisis Management Team (PZZK) | District Crisis Management Center (PCZK) |
| Communal | Community (Mayor) | Communal Crisis Management Team (GZZK) | Communal Crisis Management Center (GCZK) (optional) |

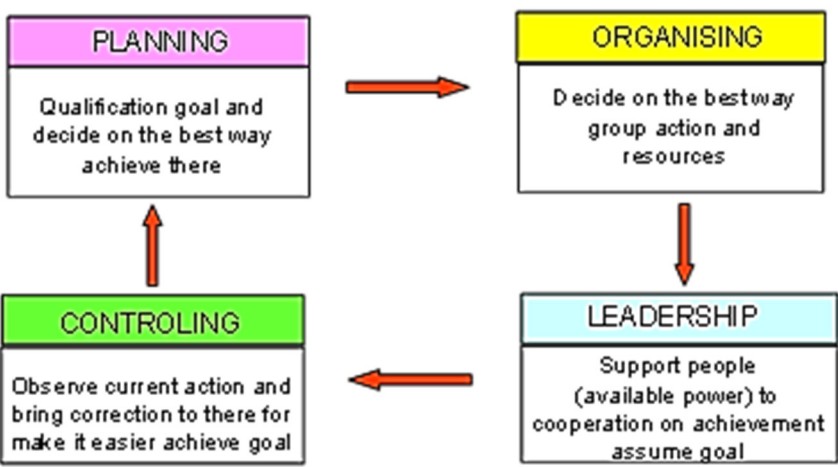

**Figure 2.** Crisis management process.

Studies of the readiness of the Polish crisis management system have revealed many serious planning errors which in the event of a long-lasting power outage over a large area may lead to serious dysfunction or even total collapse of the system under investigation. The main vulnerabilities of the Polish crisis management system faced with the growing risk of a long-term, large-scale blackout include: (1) inadequate risk assessment (underestimation of the probability and inadequate perception of the consequences of a blackout), (2) misjudgement of operational capabilities and errors in planning, (3) lack of flexibility and shortcomings at the coordination level, (4) errors in the communication of risks, and (5) exclusivity and underutilisation of collective security assets.

*3.1. Miscalculation of the Risk of Widespread, Long-Lasting Blackout*

All of the disaster management plans examined found that catastrophic power failures were incorrectly classified in the wrong risk category (tolerable, medium risk); the very same category in which local, short-term power outages are classified. The likelihood of such events and the severity of their consequences are grossly underestimated both in the actual National Crisis Management Plan [36,37] and in the crisis management plans that were developed at lower levels of public administration [38–40], as these treated the National Plan as a reference document.

The authors of the latest National Security Strategy of the Republic of Poland [35] devoted a lot of attention to energy security issues, especially in the context of the growing energy dependence of Poland and other Central European countries on the Russian Federation, posing a threat of political pressure. The document also refers to the serious

consequences of energy decarbonisation for the competitiveness of domestic electricity production, as well as the worrying condition of the generation and transmission infrastructures in the power sector, resulting from aging processes and modernisation delays related to the anticipation of new demands associated with transformation [35] (p. 8). Concerns about the growing risk of a blackout have not, however, been explicitly articulated and readiness to respond to such emergencies has not been included among the security policy priorities for the coming years, although Pillar II, point 2.7, mentions increasing resilience to energy supply threats, the ability to cope with mass events, as well as the need to collect, protect and manage food and water resources and build a resilient telecommunications and ICT system, and further the need to implement a new model of critical infrastructure protection based on business continuity [35] (p. 16), while in Pillar IV, Section 4.1. the need to expand and modernise generation capacity and the electricity transmission and distribution networks to ensure the continuity of supply and prevent unexpected interruptions in the supply of electricity, as well as to adapt the National Power System (NPS) to the operating patterns of alternative, distributed and unstable energy sources [35] (p. 34), is discussed. In accordance with the new paradigm of integrated national security management, based on the principle of seamless progression from a state of peace to a state of crisis and a state of war, flexibility, inter-ministerial and inter-sectoral cooperation and supra-ministerial coordination—the legal and administrative obstacles to increasing the military's involvement in civilian anti-crisis operations—are disappearing. This is to be facilitated by the military's increasing ability to operate in a multi-domain operational environment, increasing mobility and the effectiveness of support and logistical provision systems, enabling prolonged deployments outside permanent base locations (Sections 3.3 and 3.4) [35] (p. 18).

In the latest 2020 National Emergency Management Plan [36,37], disruptions to the energy system are included in a catalogue of 19 major non-military threats to national security, of which only flooding is classified as a high-risk threat. Blackout—defined as a sudden loss of voltage in the National Power System (NPS) over a significant area following a sequence of several random events (network failures, power plant outages, extreme weather conditions, a terrorist incident) causing an overshoot of critical values of the basic technical parameters of NPS operation (frequency, voltage) and resulting in an automatic disconnection of system components in a given area from the NPS power grid—was treated jointly with operational events, as a result of which a part of the National Power System (NPS) is out of synchronous operation with a value of either more than 5% of the current power demand in the NPS (system failures) or less than 5% of such demand (network failures) and ordinary power deficits, as sudden failures of the power grid caused by spontaneous damage to network components, the actions of third parties or the impact of weather factors [37] (p. 13). Such events were considered to be medium-risk emergencies, categorised as 3D in the 25-field risk matrix (possible events of high national security significance). This implicitly assumes the ability of the crisis management system to deal with such situations. It seems, however, that the social consequences of such events have been trivialised in the document, as they have been reduced to threats to the life and health of individuals as a result of the failure to maintain the operation of medical equipment and the threat of panic and public order disturbances. The inclusion of a sudden, prolonged, large-scale blackout as a medium-risk crisis event is, therefore, incomprehensible. The paper appropriately assesses the impact of power system disruptions on the economy, property and infrastructure. The dangerous consequences of power system failures include disruptions in the functioning of hydrotechnical equipment and communal infrastructure (sewage pumping stations, drinking water intakes, treatment stations and water supply systems), disruptions in transport (city, railway and air transport), as well as traffic chaos caused by the failure of traffic lights and problems with fuel distribution, shutdown of industrial plants and the malfunctioning of fire protection and antipollution systems, disruptions in the functioning of critical infrastructure (e.g., rescue services, telecommunications and ICT systems, radio communication, payment systems/e-banking) and the continuity of public

administration, in particular public safety and order services (non-functioning of central registry systems and restrictions in access to databases, suspension of border controls). All of these hazards, when considered in isolation from each other, were classified as medium risk and were considered to be situations that the crisis management system can cope with. The assessment of operational capacity and resource sufficiency insufficiently takes into account the combinatorial, cumulative and systemic nature of the contingencies under consideration, which occur simultaneously in the aftermath of a prolonged, large-scale blackout, producing complex structures of interactions and undesirable, unpredictable synergies. The very fact that dealing with such coupled emergencies is not possible without extensive military involvement demonstrates their extraordinariness and supports the designation of severe, prolonged, large-scale power system failures as high-risk threats. The inclusion of blackout in the medium risk category is a sign of inconsistency, considering that electricity systems are given the highest priority in security policy for the restoration of critical infrastructure. Elsewhere in the document, the authors identify electricity as the most critical resource for government, industry and domestic households. Due to the extent to which the economy and households are dependent on electricity, even a short interruption in its supply can cause serious damage, particularly in large conurbations. The authors of the plan argue that due to the characteristic features of electricity that distinguish the security requirements of electricity systems from those of other energy systems (gas, heat or fuel supply systems), such as the limited capacity to store the quantities necessary to meet even short-term customer demand and the lack of inertia resulting from the extremely short time between generation and consumption, the electricity system does not have the capacity to compensate for supply interruptions without noticeable consequences for customers [37] (p. 205). It is therefore all the more surprising to put blackout in the same risk category as failures of communications systems, ICT systems or other supply systems, which are an inevitable consequence of blackout.

The same mistake of underestimation is reproduced in crisis management plans by lower-level public administrations. The Poznań District Crisis Management Centre—one of the exemplary lower-level crisis management services operating in Poland—qualifies blackouts to the medium-risk event category 4B (probable events of low severity) and reduces the social nuisance of such crisis situations to "a negative impact on the comfort of life" [40] (p. 66).

The underestimation of the probability of a long-term blackout, reproduced in the planning documents of crisis management services at the level of local government units, results from limiting the analysis to historical data and not taking into account the conditions discussed above, which are increasingly conducive to the occurrence of sudden systemic failures. On the other hand, the underestimation of the effects and their severity is mainly due to the too low resolution of the hazard analysis, which does not sufficiently take into account the situational determinants of the severity of long-term power shortages, as well as secondary effects, system effects (e.g., synergies, feedback loops), complex multi-level interactions and crossover interactions, cascades, domino-effects and cumulations of critical disturbances, as well as many other complex cause-effect relationships [43]. Meanwhile, the severity of the social consequences resulting from a power outage and the needs for emergency responses are strongly dependent on the duration of the outage, the time of year, the degree of urbanisation of the affected area, its spatial extent, natural, socio-economic and socio-cultural conditions, days of the week, as well as many other, less important factors [25] (p. 168).

### 3.1.1. Controversies Surrounding the Assessment of the Probability of a Catastrophic Large-Scale Blackout under Polish Conditions

In view of the structural changes in electricity supply systems, most experts specialising in the design of power systems and the study of failures in such systems estimate the probability of a large-scale blackout lasting even a few days in Europe in the next five years as close to 100% [1]. A much bigger problem than the possibility of a catastrophic blackout itself is the widespread underestimation of the threat, which results in European societies

being completely unprepared for potentially apocalyptic crisis situations. The widespread downplaying of the threat is fostered by the lack of existing experience of such events. The absence of large-scale system failures in the territory of Poland in the past and the reliable operation of the electricity supply systems on a daily basis give decision-makers and the majority of the public a false sense of security that changing conditions in the operation of the electricity system will not have a significant impact on the level of energy security.

Poland has indeed had no experience of prolonged, widespread and catastrophic power system failures in the last quarter of a century, although in the past the country has repeatedly experienced smaller-scale failures combined with natural disasters and cascading outages due to overloading of generation systems. In Poland, there are several hundred cases of unplanned power outages every year. Almost 700 such incidents were reported in 2006 alone [10] (p. 681). The largest electricity blackout in the modern history of Poland occurred on the night of 7–8 April 2008 in a part of the West Pomeranian Province as a result of storm winds and heavy wet snowfall, which caused extensive damage to transmission lines supplying power to the Szczecin agglomeration and adjacent municipalities. A cascade of outages of four extra-high voltage lines supplying the left bank part of the city was triggered on 7 April 2008 at 21:25 by the automatic disconnection of the 110 kV Recław-Moracz-Goleniów line, caused by the breaking of the arms of the power pylons. In the following hours, under the influence of the breakage of insulators and lightning conductors, there was an independent shutdown of the 220 kV line, Krajnik-Glinki. The blackout left more than half a million people without electricity for several hours, completely paralysing life in the city and causing chaos and serious problems in the functioning of the region due to the damage and temporary shutdowns of many network infrastructures (including the district heating system), transport disruptions, as well as the suspension of trade, banking, production and interruptions in the work of most state institutions. Documented losses resulting from the failure in the Zachodniopomorskie region amounted to over 50 million PLN, and the authors of the report containing the results of the investigation into the causes and effects of the energy disaster have shown that, with the existing system of power supply lines supplying the agglomeration and the region, full security of power supply cannot be guaranteed in a situation of repeated extreme weather phenomena [44]. The longest lasting power grid failure occurred in January 2010 in the northern part of the Silesian province as a result of extensive damage to power lines caused by adverse weather conditions (heavy wet snowfall). The failure, which lasted for nine days locally (e.g., in the district of Myszków), deprived almost 80,000 people of electricity and heating in the coldest period of the year, forcing the local authorities to declare a state of natural disaster. The last time there was a serious blackout threat in Poland was in the aftermath of storm Xavier, which, on 4–10 December 2013, severely damaged transmission lines in several provinces and caused dozens of outages that deprived more than half a million customers of electricity. In most cases, the emergency management services of districts with traumatic experiences of local blackouts in the past assign blackouts to a higher risk category in the local risk analysis than central and local government services that have not faced similar crises before.

Underlying the widespread disbelief in the possibility of a sudden occurrence of a prolonged, large-scale blackout is a widespread perception of risk based on unreliable statistical reasoning that tends to bias the expected value of the probability of an undesirable event downwards with each instance in which the event does not occur. The fallacy of such reasoning is often illustrated by the example of the turkey, whose every successive feeding reinforces a false sense of security and trust in the good intentions of the farmer. Meanwhile, the experience of other countries with catastrophic blackouts justifies growing concern over the possibility of a catastrophic, long-term blackout in Poland and a large area of Europe, as do analyses of the contemporary conditions for the operation of electricity power systems, which reveal surprising constellations of factors that are extremely conducive to widespread, catastrophic failures.

Between 1965 and 2017, there were 138 major power system failures worldwide [45] (pp. 413–414). In Europe, blackouts in recent years have not been as severe as on other continents (e.g., north-eastern US states and part of Canada, 14 August 2003, duration: three days, 50 million affected, cause: human error initiating cascading overloads on transmission lines; Brasilia, 4 February 2011, duration: 16 h, 53 million affected, causes: transmission line failure and power flow fluctuations; India, 30 July 2012, duration: 15 h, locally even two days, 620 million affected, cause: transmission line overload; Bangladesh, 1 November 2014, duration: 24 h, 150 million affected, cause: failure of HVDC station; Australia, 28 September 2016, duration: 16 h, 21 million affected, cause: cascading failure in transmission system; Azerbaijan, 3 July 2018, duration: 24 h, 8 million affected, cause: extreme high temperatures) [46] (p. 11). The Indian blackout of 2012 has to date been record-breaking in terms of the number of people affected. In Europe, the last major sustained, large-scale system failure occurred in Italy on Sunday morning, 28 September 2003. A total of 56 million people fell within the range of the outage, which, as a result of a cascade of disturbances, covered the entire country in a short period of time and lasted about 18 h in many regions [25] (pp. 173–176). If such an outage had occurred on a working day and at a different time of year, the scale of the economic losses would certainly not have topped out at EUR 120 million, as happened with the Italian blackout. The widespread outage in Italy was preceded by a smaller one in southern Scandinavia on 23 September 2003. The blackout left about four million people in parts of Sweden and Denmark without electricity for more than 6 h. The longest power outages across Europe have so far occurred in the Slovenian region of Notranjska (outage started on 1 February 2014, cause: physical destruction of transmission lines due to severe icing, outage duration—almost two weeks) [28,29], in the German Münsterland (6 November 2005, cause: breaking of transmission lines due to heavy wet snowfall, outage duration— almost a week) and in the Berlin districts of Treptow and Köpenick (19 February 2017, cause: cascade of outages caused by damage to a cable transmission line by an excavator, duration—31 h, number of affected: about 100,000) [33]. An instructive experience of a new kind of blackout risk caused by a malicious cyber-attack, is the outage that occurred on Wednesday 23 December 2015 in Western Ukraine. Data injection (FDI) falsifying the transmission parameter measurements provided by the substation equipment condition monitoring systems led to a system collapse over a large area, depriving approximately 700,000 people of electricity for about 6 h [47] (p. 430), [30–32].

Much more frequent than outright blackouts are quasi-blackouts or near-blackouts in interconnected power systems—critical incidents threatening system collapse that are successfully contained by sheer good luck. An example of such a critical incident was the failure that occurred on 8 January 2021 at 13:04 CET in the European interconnected continental electricity networks [48]; this was the second time this had happened, following a similar incident on 4 November 2006. As a consequence of a minor fault in the Croatian network, there was a sudden and dangerous drop in frequency to 49.742 Hz, which caused the European transmission system to split spontaneously into two islands. Thanks to the swift intervention of European grid operators and the availability of sufficient temporary reserve capacity in the French electricity system, the situation was brought under control within seconds and a stable frequency of 50 Hz was restored, preventing a cascade of disconnections from the grid that would have had far-reaching consequences. After 53 s, the separated systems were reconnected, which, contrary to expectations, was managed without a hitch, despite the fact that simulation exercises conducted by transmission network operators regularly show that the reconnection of island networks often leads to a failure of the entire system.

Conditions conducive to blackouts are formed by the constellation of conditions that encompass:

- technical-structural factors (e.g., dysfunctions in the energy transmission or generation systems, including disruptions in the supply of raw material, failures of IT systems in

controlling the energy flow, a cumulative increase in the structural complexity of the power system responsible for an increase in system risk [49–53];
- natural factors (e.g., seismic phenomena, extreme atmospheric phenomena, cosmic phenomena—e.g., magnetic storms, etc.);
- economic factors (e.g., marketisation and progressive liberalisation of the energy sector, unbundling of production, transmission and distribution of energy to strengthen competition and lower prices, lack of necessary investment from the point of view of security and the growing demand for electricity, unprofitability of raw material extraction);
- political factors (e.g., international obligations limiting the ability to operate old generation infrastructures, court orders stopping the operation of mines or power plants, embargoes, trade wars, malicious cyber-attacks, armed conflicts and hybrid wars);
- social factors (e.g., conflicts with environmental organisations or local communities opposed to the construction of new power plants or transmission infrastructures) [11] (p. 671).

The disruptive impact of the energy generation on the security and stability of the electricity supply system deserves special attention. A stable energy supply only exists when there is a balance between the amount of energy produced and the amount of energy needed, and when there is a smooth flow of energy. The larger and more structurally complex interconnected systems are, the more problematic it is to balance them properly. Since non-renewable energy sources began to replace stable generation systems based on large power stations provided with raw material security—and with it, the possibility of unlimited primary energy storage—power fluctuations have appeared in the system, which, in view of the inadequacy of energy storage capacity, limit the possibility of regulating energy production. Overcapacity appears as suddenly as undercapacity—prematurely or later than originally forecast. As fluctuating power reserves shrink dangerously with the shutdown of large power plants, many experts are concerned about whether temporary demand and supply can be successfully balanced in the near and distant future. An additional problem associated with energy production is the change in the spatial distribution of electricity generation systems. Until now, conventional power plants have been located at the point of greatest demand, associated with the concentration of industries, especially energy-intensive industries, such as steel. Traditional industry has historically been concentrated around fossil fuel extraction sites, creating a symbiosis between the growth of industry and the development of energy generation in large coalfields and avoiding many of today's problems of safely transporting energy over long distances. In contrast, new RES-based systems, especially wind power, are located far from the areas of greatest energy consumption, requiring the costly development and maintenance of transmission infrastructures that are disruptive to the environment and the population, and pose numerous safety and reliability issues. An example of the spatial mismatch between energy production and consumption systems can be seen in Germany, in which the largest wind power capacities have been installed in the northern part of the country, where the population and industrial concentration are lowest. By abandoning conventional power plants, the so-called momentary reserve in the form of spinning reserves is lost; this is an important element in compensating for sudden increases in demand or losses of power in the power system, and thus stabilising the operation of the entire system. As yet, there are no market-based tools to replace this important compensatory function of conventional power plants [1].

When discussing the likelihood of a catastrophic blackout in the context of crisis management, there is no point in arguing about how likely it is that a sudden, prolonged blackout will last in a particular area, because under conditions of absolute dependence of the average European on supplies and assistance from outside, any large-scale disaster— be it extreme weather phenomena, such as catastrophic floods, massive snowstorms or earthquakes, or nuclear disasters, or even pandemics such as SARS-CoV-2—can initiate cascades of disruptive events, threatening to shut down existentially important systems and collapse key supply chains on which people's ability to survive depends. Under the

influence of secondary threats that can grow out of supply chain problems, such crises can escalate at a surprisingly fast pace into humanitarian disasters [2] that even the best configured and most efficient crisis management systems will not be able to cope with. No country is or can be prepared for such situations.

### 3.1.2. Controversies Linked to the Prediction of the Effects of Long-Term, Large-Scale Blackouts

The lack of personal experience with blackouts also limits the ability to project such emergencies and anticipate the full spectrum of their possible consequences. The widespread trivialisation of the threat is rooted in a lack of awareness of how the domino effect of harmful events that a blackout initiates in systems that are hopelessly dependent on power continuity will be far more damaging than the initial effects of a power outage. Cascading system failures that depend on electricity will certainly do far more damage than a lack of electricity alone.

Failures in servers and communications systems will quickly lead to destabilisation and paralysis in all areas of social functioning, causing, not only widespread shock, chaos and outbreaks of panic, but also, in many places, permanent and irreversible physical damage. Without connectivity, there can be no information flows, no billing, no work, no administration, no supply and distribution of goods (e.g., medicines and fuels), no coordination and no real remote working, to which humanity has switched in many areas under the impact of the COVID-19 pandemic. As there is no experience of such emergencies to date, no one knows in which areas and for how long it will be possible to maintain continuity of operation and how long it will take to restore all essential functions once the power system has returned to normal, stable operation. In many areas and sectors, it is probable that there will be no full return to the normal state and functioning prior to the disaster due to the irreversible damage or losses incurred. In many countries, there have already been trials of broad-based, systematic research into the effects of different blackout scenarios and the needs for building societal resilience and the preparedness requirements of the emergency management system to cope with such situations [17,19], although no one has yet succeeded in drawing up a complete catalogue of the possible consequences of a sudden, prolonged power failure covering a large area of the European continent. Past experience with similar situations of smaller magnitude warrants the recognition of such scenarios as a specific type of threat, not comparable to any of the elemental threats for which most countries' crisis management systems are prepared.

Experts agree that, due to the complexity and interdependence of the sectors included in critical infrastructures (energy, transport, communications and IT, health, water and food supply, finance and insurance, state and public administration, media and culture), prolonged power cuts may have similar consequences to pandemics and nuclear disasters. The consequences of such events could be the simultaneous paralysis of many important activities, spatio-temporally unlimited cascades of damage with high escalation potential, and severe disruption of supply chains threatening humanitarian disasters of massive proportions.

On the basis of other countries' previous experience with long-term, large-scale blackouts and existing scientific studies, we have identified the following potential effects of blackouts, determining the network nature of emergencies and making it difficult to maintain control over them:

- For the population: direct threat to the life and health of individuals (e.g., lack of ability to maintain functioning medical equipment, limitation of activities of hospitals and other health care facilities for the most urgent life-saving treatments, problems with access to medicines, drinking water and food, increase in the risk of food poisoning as a result of consumption of food spoilt by failure of refrigeration equipment, breakdown of communication preventing the summoning of help in life-threatening situations, disruption of social protection services and heating equipment failures which may increase the risk of loss of life or limb for the elderly, sick or disabled), panic among the population, secondary threats to public safety and order resulting from

failures of safety systems (monitoring, personal identification systems, inaccessibility of electronic databases and processing systems, alarm systems, etc.), disruptions to the functioning of public safety and security authorities, interference with the functioning of security and law enforcement agencies, increased criminality, growing social unrest, conflict and unrest, as well as problems with supplies and negative impacts on household living conditions and the plight of people stranded outside their homes (e.g., tourists) or trapped in dark and confusing transport vehicles in places lacking basic infrastructure or that are difficult to access for self-rescue, evacuation or first aid (tunnels, underground railways, lifts, cable cars, etc.).

- For the economy, property and infrastructure: failure of electronic systems used in banking, commerce and logistics (e.g., money transfers, electronic banking, cryptocurrencies, stock exchanges, commodity and currency exchanges, online sales, payment systems, cash systems, fiscal supervision, etc.), disruptions to the functioning of hydraulic facilities and municipal infrastructure (sewage pumping stations, water treatment plants and water supply networks), disruption or complete interruption of production in industrial plants operating on a sustained basis, failure of IT systems, restrictions to information flow on the Internet, risk of data loss, damage to equipment, files or software, interruptions or complete suspension of broadcasting among radio and television broadcasters, disruption of communication infrastructure—mainly mobile and fixed-line telephony, due to limited battery life in emergency power supply systems and overloads resulting from increased user activity related to reporting failures, calling for help, contacting relatives and trying to obtain information about the expected duration of the failure, recommendations of services and possibilities of obtaining assistance in time of need—disruptions in the functioning of public administration, disruptions in the functioning of passenger and freight transport—mainly rail transport and the electrified part of urban transport (mainly metro, trams, urban rail, electric buses), but also air transport, due to the suspension of departures caused by the failure of electronic data processing systems and failures of safety infrastructures deprived of backup power sources—failures of traffic lights contributing to traffic chaos, an increase in the risk of collisions, accidents and catastrophes in terrestrial traffic and the creation of numerous bottlenecks impeding the efficient movement of emergency, rescue and law enforcement services;
- For the environment: local contamination of the environment as a result of, inter alia, disruptions to municipal infrastructure (sewage treatment plants and pumping stations), disruptions of operations at plants storing or producing harmful or hazardous chemicals—including Toxic Industrial Chemicals (TICs)—threatening their release and failures of hydrotechnical facilities used for damming or diverting water, as well as deliberate pollution of the environment caused by failures of environmental monitoring systems and disruptions in the operation of environmental surveillance services.

The alarming catastrophic potential of widespread and prolonged blackouts is clearly evidenced by the cost and economic loss assessments drawn up after the catastrophic blackout which, on 14 August 2003, left some 60 million people in north-western US states and parts of the Canadian province of Ontario without electricity for over 20 h. A cascade of more than 100 power plants being disconnected from the electricity grid due to a string of transmission line overloads caused severe losses in the US economy, estimated at between $4.5 bn and $8.2 bn. The balance of losses, however, takes into account only the loss of revenue of the companies prevented from operating (US$3.12–5.20 billion), losses resulting from the destruction or perishable goods (up to US$1 billion), losses for energy generators and suppliers caused by more than a twenty-hour interruption in the sale of energy to consumers, primarily to the industrial sector (US$1–2 billion), and losses in taxes [23] (p. 6). The approximate value of damage and distributed, uncatalogued losses has not yet been comprehensively estimated, but it can be assumed that the actual economic losses caused by the accident were disproportionately higher than those included in official statistics, while social and environmental damage would be even more difficult to quantify [10] (p. 683).

As a result of a sequence of failings, a long-term, large-scale blackout in Poland is still—despite the recommendations of experts—considered in a generalised way as a threat defined as an 'electricity blackout' together with events of low or moderate social harmfulness, such as planned outages, local failures or short-term power cuts. From the point of view of the consequences and challenges for crisis management, such events have nothing significant in common, which precludes their being considered together. The failure to properly identify the possible effects of a prolonged, large-scale blackout and the gross underestimation of their severity result from the adoption of a reductionist cognitive perspective based on mechanistic thinking that seeks to understand complex realities through their elementarisation and reduction to simple, quantifiable deterministic-linear causal relationships [53] (pp. 82–83). Such a view systematically loses sight of both the "dangerous links"—structural couplings and interactions between individual hazards, cross-influences, synergies, mutual reinforcement, cumulative and self-organising capacities—and non-measurable impacts (e.g., widespread damage). The consequence of underestimating the risk of a sustained large-scale blackout is a false sense of security and consequent planning errors [43].

*3.2. Errors in Operational Planning*

Excessive elementarisation of risks, while ignoring the complex causal relationships between them, results in a failure to develop suitable response strategies. Individual risks are dealt with separately by planners and countermeasures are planned for each risk separately. Consequently, real-world events, such as a sudden, prolonged, large-scale blackout—causing knock-on effects and cascades of network disruption resulting in complex, multi-level, cumulative emergencies—torpedo any emergency response plans. Faced with real blackouts and the sense of loss of control typical of such situations, the need to secure already insufficient capacity and resources, combined with a misguided, over-optimistic assessment of one's own operational capabilities, resulting from inadequate theoretical and practical preparation (lack of simulation exercises) will further compound surprise, shock and trauma, and will certainly not facilitate the rapid development of alternative response options and an agile transition to a contingency mode more appropriate to the situation during the first hours of a blackout, which are, in fact, crucial to the success of the entire anti-crisis operation. Good judgement and the decisions taken in the first hours of a blackout, when emergency power systems are still sustaining the basic functions of critical infrastructures, will determine the success of most emergency responses implemented under the conditions of cascading failures, once almost everything has failed.

Incorrect planning of measures to deal with a long-term, large-scale blackout is due, on the one hand, to over-optimistic assumptions that infrastructures of critical importance for crisis management will not fail, primarily: electronic databases, electronic data processing and remote access systems, communication systems, emergency notification and public alarm systems (1), strategic reserves (2), logistics and transport (3), and that shock, uncertainty and critical household shortages will not cause widespread absenteeism in the emergency services, and that secondary threats will not grow on the "fertile ground" of a crisis situation (e.g., outbursts of panic, social conflicts, riots and mass looting, increases in common crime under conditions of prevailing darkness, non-functioning of video surveillance and security engineering systems, chaos, disruptions in the functioning of security bodies and law enforcement services and police preoccupation with crisis activities, terrorist attacks, industrial disasters, etc.) and that the international situation and defence requirements will allow for unlimited use of the military. In practice, such conditions are rarely met (an example of a "systemic" effect is the paralysis of transport: interruption of rail transport—trains, metros, trams—results in increased road traffic, which, in conditions of inoperative traffic lights, leads to total traffic chaos in cities and traffic gridlock. Under such conditions, the assumption of being able to effectively move the assets and resources (e.g., generators, fuel supply, distribution of humanitarian aid, etc.) needed to deal with the

crisis and reconstruction is utopian. On the other hand, the requirements for responding to emergencies caused by a sudden, prolonged and large-scale blackout were defined in the surveyed documents in a too one-size-fits-all, single-option manner, without taking into account the various emergency scenarios that depend on variables, such as the scope and duration of the emergency, the time of year (length of day, air temperature, availability of food based on growing season, no impediments to transport and construction works, etc.), the degree of urbanisation and the number, density and distribution of the population, environmental conditions (thermal conditions, topography, etc.), and economic (wealth of municipalities and population, etc.), technological (infrastructure, degree of dependence on electricity, prevalence of photovoltaics and possession of alternative power sources, etc.), socio-demographic (age structure and health status of the population, size and resources of households, housing conditions, household wealth, possession of own water sources and home grown or farmed products etc.), and socio-cultural (organisational capacity, level of knowledge and intelligence, willingness to cooperate and support each other, active citizenship, good-neighbourly relations, charity, stockpiling habits etc) characteristics. The risk assessment that forms the basis of emergency planning is too rigid, treating the severity of the failure and the level of difficulty in dealing with such situations as a fixed quantity. However, in the case of a blackout, the level of risk and challenges to crisis management depend on complex local conditions and circumstances, and above all on the duration of the failure, which is difficult to predict at the time of risk assessment and decision-making crucial to the success of subsequent emergency operations. Due to the different levels of resilience of urban and rural populations to supply crises, it is not possible to establish uniform, universal critical thresholds defining the catastrophic levels of blackout and differentiating emergency response methodologies depending on the duration of the failure. As a system of measurement, one could adopt time frames determined by the percentage of the population that are unable to survive independently in a situation of widespread blackout without external assistance—with individual time frames for different degrees of urbanisation. For large urban agglomerations, such a critical threshold would be set at three days. This is, on the one hand, the legal minimum time for critical infrastructures to remain operational and, on the other hand, the time after which more than a third of the population would need external assistance to survive, above all for the supply of basic necessities, which would overload the social assistance and civil protection system and make it necessary to call for external assistance. The second critical threshold for the duration of a blackout in a metropolitan setting would be a week, since under such conditions less than a third of the population would be able to survive for hours or days without outside assistance, leading, in practice, to the total failure of the civil protection and humanitarian aid system. In the event of a spatially widespread accident, the involvement of the military would be unavoidable. For rural areas, the proposed critical thresholds should be doubled (from three days to one week and from one week to two weeks) due to lower population densities and greater self-sufficiency. However, an analysis of the planning documents produced by crisis management centres of large cities and rural municipalities did not reveal any significant differences in blackout risk assessments and response methodologies. Different alert levels and different procedures have not been defined for various states of emergency, as is the case with terrorist threats, for example. Yet, risk assessment and response requirements should be dynamic, flexible, adapted to changes in threat levels.

### 3.3. Insufficiency of Crisis Management Resources

In spite of warnings from experts, in Poland—as in most other countries—the risk of long-term, large-scale interruptions to electricity supplies is downplayed or relativised. No convincing and workable concepts for overcoming such cumulative networked crises have been proposed to date. As such situations have not yet occurred on a larger scale in Poland, there is a lack of traumatic personal experience of such dramatic situations, which favours the suppression of this problem from public awareness and treating the warnings of experts, who demand urgent spending of large sums on building emergency

power supply systems and preparing for such situations, with incredulity. Poland—like most industrialised countries—has an increasingly limited capacity to respond to this type of crisis. This is mainly due to public sector indebtedness and the need for radical austerity, privatisation, outsourcing and increasingly scarce strategic reserves, reduced due to the influence of modern just-in-time business models. All of this adversely affects the maintenance of disaster preparedness and response capacity in disaster or emergency situations, emergency planning, resource management, etc. In the event of a regional or national blackout, increasingly lean emergency management systems can rely on external support. However, in the event of an international blackout, there would be very limited scope for mutual assistance. If individual states had to rely only on themselves in such crisis situations, the response capacity would, in many cases, prove inadequate even with the increased engagement of the armed forces. Preparing effectively for sudden, cumulative emergencies with a high potential for escalation is essential, because when such situations occur, everything that has not been thought of before will be lacking and there will no longer be time to organise the necessary resourcing. Organisations responsible for crisis management need to know at what point and how quickly to switch into emergency mode, exactly what needs to be done to do this and be prepared for all possible situations. They must be able to adequately assess the situation, to determine—at least approximately—how long a given crisis situation will last, what the requirements will be during that time, what resources are available and for how long these resources will last, how to make optimal use of the resources available and what to do when they run out. The preparation of logistics facilities, including the supply of equipment and other resources in appropriate quantity and quality, as well as the concept of their deployment—taking into account crisis conditions—should be underpinned by a concept of strategic crisis management in case the assumptions and strategies adopted turn out to be inadequate and the countermeasures undertaken fail.

It is the task of the authorities and services responsible for crisis management, security and civil protection to organise headquarters, information and contact points and help desks, which should be as evenly distributed as possible so that every resident has these within one hour's walk, i.e., within 3 km of their place of residence. Medical assistance and basic medicines, drinking water and other necessities should be available at these locations. Longer distances would be difficult for those in need of medical assistance. In addition, carrying drinking water or food over longer distances would be onerous even for healthy people in good physical condition. In the process of emergency planning, it is necessary to foresee such organisational activities and the resources needed for this: human, property, material, etc. However, if there is no mutual (neighbourly) assistance at the level of human interaction, then in crisis situations in general, and in situations involving a long-term blackout in particular, the crisis management services can plan all sorts of top-down measures at will, which, due to staff shortages, fuel supply disruptions, etc., will only work to a limited extent or not at all. Therefore, an integral element of the crisis management system is crisis management in households and small local communities (e.g., neighbourhoods) and the success of actions taken at the central and local government levels depends on the efficiency of this most basic link.

### 3.3.1. Challenges for Crisis Management Resulting from Failures in Communication Systems

One of the main factors limiting the efficiency of complex multi-agent systems is communication and coordination problems. A prerequisite for good interoperability is reliable communication systems that can tolerate prolonged power cuts. The emergency services of most countries have digital radio communication systems similar in structure to commercial mobile phone networks, with the difference being that over short distances (up to about 1 km) it is possible to establish direct communication between individual devices. In the event of a power outage, the transmitters use batteries that provide continuous operation for only a few hours. In this situation—as in the case of operations outside the range of the home network—in practice, only short-distance direct communication remains.

To increase the range of communications, signal repeaters are used, but they do not solve all of the communication problems between emergency headquarters, command centres of individual services and operational units during operations conducted over large areas for several days. During the emergency response in Slovenia in February 2014, communication was supported by the amateur shortwave communication network "Win Link" [29]. By the time most rescue units arrived at the scene of the disaster, the power cut had already lasted for more than 24 h, so the signal on commercial mobile networks had disappeared due to the lack of power. One mobile phone operator only managed to secure emergency power on the fourth day of the blackout, but its transmitters only broadcast a low power signal. A good quality connection was only possible to achieve outdoors and in the open. Twenty special mobile phones with increased transmitter power were made available free of charge by the operator to the emergency services, but their batteries only lasted a few minutes and their operation was quite complicated. A much better solution would have been to briefly connect the emergency services' private mobile phones to the network for the duration of emergency operations, but the network operator did not agree to this for fear of abuse. For overseas rescue missions, communication with national command centres is often only possible using satellite communications, which provide stable and efficient connections. However, the most effective communication is arguably via the Internet. The installation of WLAN routers at emergency sites allows a larger group of individuals unlimited access to information resources, messengers and exchange of information via social networks. Wireless technologies are key, enabling efficient communication and information transfer over long distances with minimal electricity requirements. Because emergency power supply battery buffers are time limited, in the event of a blackout, telephones, mobile phone relay stations, internet infrastructure and service radios, including some military communication systems, usually go dead after a few hours. Emergency planning should therefore take into account alternative communication concepts in the event of prolonged, multi-day power cuts.

3.3.2. Securing Emergency Power Supplies for Emergency Headquarters, and Facilities and Infrastructure of Key Importance for the Success of Emergency Operations

Driven by the poor communication of risk by the crisis management services, the public in Poland expects the state crisis management services to have a sufficient number of power generators, which are in full readiness for immediate use whenever this proves necessary. However, an analysis of the crisis management plans drawn up by districts and municipalities leads to the sad conclusion that administrative bodies do not have adequate knowledge of the need for emergency power supply, not only in relation to the most densely populated areas or those where there are critical infrastructure facilities or facilities requiring special protection (hospitals, hospices, social welfare homes, prisons, etc.), but even with respect to their own needs. The emergency services have, at best, one emergency power supply device to meet their own needs and which serves only to maintain their day-to-day operational capacity. A research team from Witten-Herdecke University, developing concepts for the protection of KRITIS critical infrastructure under conditions of prolonged, large-scale blackouts, estimated the average minimum electricity demand of a municipal or district emergency headquarters at around 50 kWh. Basic equipment for an emergency headquarters included several laptops, a desktop computer, two printers, mobile phone chargers and torches, plus lighting for the headquarters office, entrance, corridor and toilet, and optional heating equipment. This leads to the conclusion that every emergency management centre should be equipped with an operational generator with a minimum capacity of 60 kVA [54].

The limited number of back-up power sources—including high-powered mobile generators—located in state and local government reserve storage facilities allows only localised solutions to be available. The sense of security they generate under conditions of a prolonged, large-scale blackout will prove illusory, with potentially dire consequences—for example, adding to the shock caused by the sudden paralysis of almost all technical infrastructures. For normal power supply, an average electricity demand of 1 kWh per

person can be assumed. Even if the average demand in the event of an emergency power supply is assumed to be one third of the normal demand per person, in the event of a large-scale blackout, the demand of a large population would be difficult to meet using the capacity of the available back-up generating systems. One high voltage line (110 kV) can supply 100 MW of demand. Typical emergency power supply systems in the fire brigades of most European countries have a capacity of 150 kVA. Larger gensets of 500 kVA are also used for special tasks. The total capacity of all 172 gensets delivered by international assistance to the Slovenian region of Notrjanska, where a prolonged blackout occurred in early February 2014, with a population of about 100,000 in its range, was about 20,000 kW, which proved to be far from sufficient in practice [29]. However, so far neither in Poland nor in most European countries has there been an inventory of the actual number of generators, their condition, total power and demand for fuel—equipment owned by the services integrated into the crisis management system—as part of the management of the logistic facilities for this system.

The Polish armed forces have at their disposal mobile, containerised high-power generators (e.g., 250 kVA) and synchronisers, which enable the creation of larger emergency power supply networks with the capacity for rapid relocation thanks to modern handling systems. The model 1000 kW Containerised Field Power Plant consists of four containerised modules: two generating sets with a capacity of 280 kW each, one generating set with a capacity of 440 kW and a power distribution set. The mobile power unit is diesel-powered and the nominal fuel consumption at full load is $2 \times 63 + 1 \times 80$ L/h, so a total of 206 L/h, resulting in a daily fuel requirement of 5000 L. The generator set has three fuel tanks: two with a capacity of 2000 L and one with a capacity of 2300 L, which enables continuous operation at full capacity with a single filling for approximately 30 h. This unit, with a total combined capacity of 900 kW, can supply emergency power to a town with up to 20,000 inhabitants, at a cost of around EUR 400,000. This basic overview leaves no illusion that even the richest countries cannot afford to organise an emergency power supply system that will safeguard the needs of crisis management under conditions of a prolonged blackout over a large area. An additional problem is the limited knowledge of the emergency services regarding the efficiency of their back-up power sources. Information on the technical condition of power generators is frequently omitted from power inventories at the disposal of local agencies integrated into crisis management systems.

Therefore, having such devices does not fully protect against risks, especially if local authorities do not know which critical infrastructures will function and which will not in the event of a blackout. The public must be informed of expected concerns in good time so that they can prepare adequately for possible emergencies on their own. For the purpose of planning around the need for emergency power supply in situations of prolonged blackout, studies of the population's behaviour related to the consumption of energy from emergency power supply systems after a prolonged interruption of energy supply are also urgently needed. Excessive consumption may undermine estimates of the emergency power supply capacity needed. None of the crisis management units in Poland, as part of their planning, collect information that would allow for estimating different levels of emergency power demand in the case of crisis scenarios varying in terms of spatio-temporal coordination (e.g., spatial extent and number at risk, time of year, duration of power outage, etc.). Without such knowledge, it is difficult to start forecasting and planning for emergencies. This is because there is no way of knowing how many gensets of what capacity might be needed at what location, and what the resulting needs are for transport, operation and supply of fuel (diesel, petrol, gas) for such equipment.

In many places, simple emergency power supply systems are also being exploited by retrofitting commercial vehicles in municipal stock with more powerful generators and batteries and power inverters. Such mobile systems can be deployed on a rotational basis to the locations where they are most needed in order to supply emergency power to critical equipment, e.g., the central heating pumps of buildings connected to the municipal or district heating network, which is extremely important for survival in winter conditions.

It is sufficient to equip such devices with a suitable extension cable to connect them to a vehicle outside. Such mobile emergency power systems will render invaluable assistance in the process of damage remediation, repair and restoration of important infrastructures, especially in conditions of darkness [55].

There are, of course, institutions with their own back-up electricity supply systems, such as hospitals. However, the reality of the energy security of such facilities rarely corresponds to the documentation and legal requirements relating to them. Technical documents and safety inspection protocols for emergency power supply systems, which are mandatory for operators of critical infrastructures, provide information on the technical parameters, the total capacity of the equipment, the fuel requirements and reserves held, and the matching of capacity with the minimum electricity demand, but information on the technical condition of the generators is frequently missing. Even a cursory review of this type of equipment in Polish hospitals shows that these are older generation units, often manually activated, partly worn out and partly of unknown efficiency due to infrequent use. Few modern devices are equipped with automated systems, which turn the genset on without operator intervention immediately after the power supply fails. Starting the genset requires powering a starter motor driving the genset from a functioning battery pack, similar to starting an engine in a car. Batteries are characterised by the fact that their capacity (ability to store electricity) decreases as they age and, with old batteries, even the slightest difficulty in starting the combustion engine leads to the exhaustion of the available energy reserve. The low frequency and—usually—short service life of uninterruptible power supply systems mean that the reliability and safety requirements of such equipment and service obligations are widely neglected. Most operators of such systems do not know, for example, how many hours the engine will last on the fuel supply, or how long since the fuel tank was last filled or cleaned of solid debris and water build-up. Having your own power generators, therefore, does not guarantee full safety in blackout conditions, as one hospital in Berlin's Köpenick district painfully experienced during the recent blackout, when the generator providing the hospital with emergency power failed after only seven hours of operation and patients had to be evacuated immediately [33].

Microgeneration is becoming increasingly popular in discussions about emergency power systems as an antidote to power outages. However, more and more owners of domestic photovoltaic installations have an illusory feeling of security, as the vast majority of such devices are rigidly connected to the electricity grid and only supply electricity when the grid is functioning normally. Few devices of this type are equipped with their own energy storage and have the ability to automatically disconnect from the grid and operate as an "island". Many owners of such systems do not know this. Viable investments in generation capacity are rarely matched by unrewarding investments in storage buffers, with the result that only few prosumer systems have the capacity to operate autonomously under grid failure conditions.

In order to build a well-functioning emergency power supply system capable of coping with long-term, large-scale blackouts, an effective concept and strategy must be developed long before the occurrence of such crisis situations. The configuration of such a system is a multi-stage learning process, requiring continuous adequacy testing, refinement and maintenance. We cannot be content with the pilot studies that have been conducted so far, which are fragmented and limited in scope.

### 3.3.3. Securing the Continuity of Fuel Supply

The preparation of a multi-variant logistics plan, taking into account different crisis scenarios and possible impediments to the movement of equipment and fuel supply, should be an integral part of planning. The crisis management services should prepare plans for the use of transport and generators in crisis operations and estimate the related fuel requirements, as well as determine the minimum level of reserves and their optimum location and secure continuity of supply, taking into account possible difficulties.

For example, the predicted daily fuel consumption of 50 volunteer fire brigade vehicles with an average consumption of 25 L/100 km and a mileage of about 1000 km will total about 12,500 L, while the predicted daily fuel consumption of 20 generators operating at full capacity will total about 20,000 L [29]. Therefore, a fuel reserve covering the maximum working time needed to deliver a new supply of fuel from the nearest active filling station, preferably using mobile tankers, including the time needed to fill them (in the case of a normal dispenser, refuelling 7000 L of fuel takes about 2 h), should be stored in the area where the crisis operations to be conducted, and a suitable place to refuel vehicles should be prepared. As dispensers at civilian service stations usually have built-in delivery limits (in Austria 750 Euro), filling a tanker requires the assistance of a station attendant [29].

The storage and transport of fuel reserves requires special protection against theft. The transport of generators and fuels itself raises questions about securing the means of transport with adequate fuel reserves. This is certainly not a pseudo-problem, as the average daily fuel consumption of an average fire service unit operating in a major city is normally around 10,000 L. In a crisis situation with increased intensity of operations, the daily fuel consumption of emergency supply systems operating at full capacity, i.e., 10,000 to 20,000 L of fuel in addition, must be added to this.

Sudden crisis situations make it acutely clear how poorly thought out some practices in the emergency system are, such as the practices of replenishing material reserves. If the storage tank has a capacity of 30,000 L, for reasons of logistical overheads, it is common practice to replenish the stock only when the stock level reaches a critical value, such as 500 L. In the event of an emergency, such practices can prove potentially disastrous [55].

### 3.3.4. Strategic Material Reserves

The custom of maintaining critical material reserves for emergencies became widespread in Europe during the Cold War under the threat of nuclear conflict. A well-thought-out crisis management strategy, taking into account the requirements of preparedness for a long-term, large-scale blackout, should take into account properly assessed equipment and material needs (e.g., equipment, tools, tents, camp beds and blankets, fuels, field kitchens, food, drinking water, clothing, protective measures, medicines and dressing materials, hygiene supplies, etc.) for both emergency services and the immediate survival needs of the population, as well as a plan for the optimal distribution of such critical reserves to allow easy access and efficient distribution under conditions of major disaster, emergencies and possible transport disruption. More and more regional authorities are taking the threat of blackout more seriously and are preparing for such crisis scenarios, but isolated organisational efforts may not be enough, especially as, for many organisations, they are limited by preparing to deal only with the first phase of a blackout. Few stakeholders, however, take into account that in the event of a long-term, large-scale blackout, the full recovery of the supply of goods of fundamental importance for survival may not occur until several weeks after the failure, and therefore the resources stockpiled for such emergencies must be considered to be far from sufficient. However, there are many dispersed resources at the local level that can provide an "invisible reserve" and may prove to be indispensable for resilience and survival in long-term emergencies, including the subsequent phases of a blackout. Governments and administrations should promote initiatives in the field of local cooperation and community aid, and support the development of local, virtual food markets, which in crisis situations could prove to be the last resort for supplying the population. A particular challenge is the provision of stocks of perishables, which go to waste after only a few hours and themselves create additional problems in situations where waste disposal is not functioning [55].

### 3.4. Lack of Flexibility and Coordination Problems

In Poland, tasks related to civil protection and crisis management are assigned to all levels of public administration in accordance with the constitutional principles of self-governance and subsidiarity. The multi-layered nature of this structure and the multitude

of stakeholders integrated into it, between which there are mutual incompatibilities (e.g., at the level of subordination—authorities of joint administration vs. authorities of non-associated administration) pose serious organisational and coordination challenges to crisis management. Responsibilities for dealing with the consequences of a catastrophic blackout have been divided in Poland in a complex way between Polskie Sieci Elektroenergetyczne—the operator of the National Power System—and the organs of government and municipal administration, but in the event of a long-term, large-scale blackout, coordination of such actions at higher levels of administration is made difficult because of communication problems and insufficient resources. However, in the event of a widespread, long-lasting blackout, the competences and responsibilities of central authorities are not as clear-cut as in the case of elemental hazards and certain cascading hazards such as pandemics. Meanwhile, in view of the inadequacy of resources and inadequately planned capacities, means and procedures, the key issue is their rational, effective use, which requires good coordination of action.

At the provincial level in particular, the public administration has a complex structure, consisting, on the one hand, of bodies subordinate to the provincial governor, who is the regional representative of the government administration appointed by the Prime Minister, and, on the other hand, of self-governing bodies subordinate to the Marshal elected by the provincial assembly. According to the law, tasks related to ensuring security, civil defence and crisis management are the responsibility of services subordinate to the provincial governor. It is the duty of provincial governors to develop provincial crisis management plans, to mutually agree upon district crisis management plans with the provincial plan, to prepare alert and warning systems, to conduct training, exercises and trainings on crisis management, to educate inhabitants on how to properly prepare for crisis situations and how to behave in such situations, and to appoint and convene the provincial Crisis Management Team [37] (p. 89). In many respects, the provincial crisis management authorities acting as an intermediary between the central and local security administrations are the most critical link in the structure of the Polish crisis management system. This way of organising the crisis management system was created for traditional threats related mainly to natural disasters occurring on a local scale, which could include at most a few districts, but from the point of view of managing crisis situations resulting from a long-term, large-scale blackout or other large-scale threats, the placement of provincial-level bodies in the coordination processes between government bodies and municipal and county headquarters threatens to cause serious disruption. The crisis management system operating in Poland is a structurally and functionally complex framework. Coordinating the operation of such a complex system, especially in complex crisis situations with high escalatory dynamics, is a major challenge. In extreme cases, the curse of complexity makes security systems a greater threat to themselves than the situations they are supposed to protect society from.

The Polish crisis management system is also characterised by excessive formalisation and bureaucracy, resulting in overly long reaction times, as well as excessive routinisation and unduly rigid procedures that are unreliable in the event of unexpected situations. Dealing with rapidly escalating, multi-level emergencies with large spatial coverage under conditions of communication system failures presents decentralised emergency management systems with major challenges in coordinating interoperability. It is simply impossible to deal effectively with this kind of complex crisis situation without the multitasking assistance of the military and the extensive involvement of community-based security assets. In such conditions, an excessively complicated, multi-level, hierarchical and bureaucratic structure of decision-making and agreement, combined with a passive approach to security, limited to rigid adherence to procedures for fear of overstepping one's authority, as well as with ubiquitous routinisation, which kills creativity—so important in conditions of sudden, shocking crisis situations spiralling out of control—and manifests in, amongst other things, widespread copying of solutions adopted by other territorial local government units in Poland, including their mistakes and with no regard for differences at the local level, guar-

anteeing that to yesterday's dramatic crisis situation caused by the day before yesterday's power failure the emergency services will react tomorrow, and their reactions will have an effect the day after tomorrow. When confronted with the escalation potential of crisis situations caused by a long-term, large-scale blackout, the organisational solutions applied to date in crisis management in Poland are a recipe for disaster. It is therefore necessary to develop and try out alternative ways of interaction between all of the organisations responsible for the various domains of security, safety, rescue and the maintenance of order in conditions of shortage or lack of power and of major serial collapses in the infrastructures on which these organisations depend for their functioning, in particular command units, rapid response services, operators of critical infrastructures (e.g., communications systems, emergency services, humanitarian aid, damage control, etc.) and operators of dangerous industrial installations with a high potential for detrimental effects and a high risk of release (e.g., nuclear reactors, chemical or biotechnological installations, etc.). The mobilisation of all social protection assets, including the mobilisation of the civilian population to take active responsibility for their own security, to be self-sufficient and to be properly prepared to cope with prolonged power outages without external assistance, therefore assumes great importance. As the ability to overcome such crises requires harmonisation and good coordination between all entities dealing with security issues and involved in crisis management, all such entities should be involved in the development process as much as possible, not only the crisis management services operating in the public administration sector at various levels, but also corporate crisis units and social organisations (self-help, humanitarian, educational, etc.). In particular, electricity distribution companies should develop scenarios for the optimal use of emergency supply systems provided by emergency services, plans to integrate such equipment into the grid, prepare the necessary cabling and define priorities in supplying customers.

*3.5. Inadequate Risk Communication and Failure to Prepare Civilians to Deal with Supply Crises Independently*

Dealing with the chaos and paralysis caused by a prolonged, large-scale blackout requires not only good preparation on the part of the emergency services, but also the preparation and active involvement of the entire population. Indeed, a large part of the population cannot rely on external assistance in such situations, as the services responsible for civil safety and security and for crisis management are also affected by the blackout in various ways and have limited capacity to act. Normal procedures, e.g., for alerting the public, crisis communication and the provision of assistance, are therefore not to be expected. Under conditions of prolonged blackout, normal interventions by the police, fire brigade, army or aid organisations cannot be counted on, as all services will face absenteeism and reduced capacity, and will be overly preoccupied with helping people with reduced mobility and independence. However, exaggerated public expectations are associated with the activities of such services, reinforced by political messaging suggesting that administrative security systems are optimally configured and equipped and that they are adequately prepared for any crisis scenario.

Research to date indicates that one third of the population in Western European countries will not be able to meet the most basic biological survival needs on their own without external assistance by the fourth day of the blackout at the latest, while the remaining two thirds of the population lose their ability to survive on their own after one week. This means that, in the event of a prolonged blackout covering the whole of Poland, more than 10 million people—especially those living in large urban areas—will be unable to provide themselves with basic resources and necessities at the level required for biological survival after just three days, and the number of people needing assistance will double after a week. Due to the specific characteristics of living in large urban agglomerations, the self-sufficiency of urban populations and their ability to survive prolonged power outages and failures of electricity-dependent infrastructures are lower than those of rural populations. Past experience with blackouts confirms that surviving the first two weeks of an emergency situation is definitely more difficult for populations residing in high density

urban agglomerations than in rural areas [29]. This is due, on the one hand, to the greater dependence of the urban population on electricity to meet personal needs and, on the other hand, to the greater self-sufficiency of the rural population in terms of food, access to water, fuel or even the possibility of satisfying physiological needs in the open air. In order to ensure that those capable of self-sufficiency do not unnecessarily absorb the limited resources, energy and means available to emergency management services, which will at best be on the edge of capacity in a blackout, it is important that households are properly prepared to cope on their own with a prolonged power outage and the resulting supply problems. The longer an individual citizen or an individual family is able to meet their needs on their own, the more this relieves the emergency management system, services and humanitarian organisations, which in a blackout are permanently overstretched and, in many ways, impaired in their normal functioning, further increasing the risk of overload. Such services should not be absorbed in helping healthy, able-bodied citizens who, out of convenience or complacency, do not stockpile for emergencies. People who are well prepared to deal with different types of emergencies on their own will be a key safety buffer. The effectiveness of crisis management and the speed of the return to normality depend to a large extent on the individual foresight and thrift of the citizens, proper preparation and the ability to cope independently with crisis situations caused by a prolonged power failure, because only those who have prepared adequate emergency stocks and can survive under such conditions will be able to be involved in the recovery phase and the restoration of functions [56]. Without an appropriate level of social resilience, it will be very difficult for society to return to its pre-disaster state of normal functioning, because people who cannot cope at home will not turn up for work and there will be no one to clean up the damage, rebuild and restore the normal functioning of systems on which the survival of the population depends.

Almost four decades have passed since the times of the Polish People's Republic "PRL", which was characterised by a continuous supply crisis and taught the Polish population how to cope with various types of shortages (including food, medicines and other necessities, electricity, water, heat, etc.), and a decreasing proportion of the population has experience in dealing with various crisis situations and maintains the ability to survive under conditions of long-term power shortages and supply problems (such ability is possessed mainly by people who partake of caravanning or hiking "in the open air"). The omnipotence of the welfare state and the standards of reliability that characterise modern technology have accustomed people to living in a safe and predictable environment. Polish society—like the societies of Western Europe, after the tension caused by the end of the so-called Cold War and the postponement of the spectre of nuclear conflict between East and West—has lost its ability to switch to emergency mode. Meanwhile, preparedness for dramatic emergencies caused by a prolonged, large-scale blackout requires a profound change in the consciousness of people accustomed to operating online every day, overconfident in the reliable functioning of the technical-social-economic super-system and living with an exaggerated sense of security. It requires the preparation of a contingency plan (Plan B) to enable a smooth transition to safe offline crisis operation without unnecessary shock and associated chaos.

Proper preparation of civilians to cope on their own in conditions of a prolonged blackout is certainly not supported by the risk communication commonly practiced by crisis management services in Poland and this perpetuates a false sense of security in local communities and exaggerated expectations from the crisis management system which, in the case of large-scale, community-wide emergencies, cannot meet those expectations. Failure to properly communicate risks is partly due to the dismissive attitude of emergency management services to the threat of catastrophic blackouts. This downplaying of the significance of the threat is evidenced by the fact that even those employed in such services are often no better prepared for prolonged emergencies than the rest of society However, if the majority of citizens do not acquire the capacity for self-sufficiency that would allow them

to survive in paralysed supply chains for a period of at least two weeks without external assistance, then crisis management systems must be prepared for extreme overloads.

Despite growing awareness of the importance of social resilience to cumulative emergencies caused by long-lasting, large-scale power outages, Poland has so far failed to run extensive social information campaigns to prepare civilians to cope on their own in such situations, in which any administrative crisis management system—even the best configured and resourced—is doomed to be overloaded due to the destructive impact of power outages on the functioning of all other critical infrastructures and the huge number of victims and those in need of assistance. The dashed expectations of the population, the pervasive sense of helplessness and powerlessness, and the frustration of people who were not warned in time of the possibility of the dramatic secondary effects of a blackout and remain completely unprepared for supply shortages, and are condemned to long waits for assistance under chronic overloads of the crisis management system, further compound dissatisfaction and frustration and increase the risk of behaviour that threatens public safety and order. Addressing such secondary threats unnecessarily absorbs energy and resources that are insufficient for more urgent needs in the context of cumulative emergencies. The following principle applies in crisis management: the better prepared the population is for emergencies, the longer they can be counted on to behave positively. Therefore, one of the key objectives of preventive action is to postpone the moment when, influenced by the scarcity of resources, selfish motives prevail over attitudes of solidarity and cooperation, leading to violent social strife.

Preparing the population for emergencies caused by a prolonged, large-scale blackout requires comprehensive public information and education campaigns to make citizens aware of the seriousness of the threat, to mobilise citizens to take responsibility for themselves and their loved ones, and to equip citizens with the ability to plan for emergencies themselves and to make a smooth, safe transition into emergency mode without shock or stress, which often secondarily threatens security to a greater extent than primary supply shortages and disruptions. Such social campaigns promoting preventive and responsible attitudes towards one's own safety and honest risk communication are not facilitated by the disappearance of the social mission of mass media, which is a consequence of the change in the way content production is financed. If the media cannot be relied upon to help, government and local authorities should take matters for promoting safety culture into their own hands and initiate dialogue with stakeholders and citizens in the form of brochures and information meetings, workshops and various participatory procedures corresponding to the open government model [57]. A programme of general preparedness for a long-term, large-scale blackout should also be extended to local businesses.

Effective self-protection, self-help and neighbourhood assistance require knowledge of survivalist principles and the preparation of adequate stocks of the most necessary subsistence goods, which may not be readily available to the wider masses under blackout conditions [58]. Every citizen responsible for their own safety and the safety of their loved ones should consider what consequences a prolonged power failure may have for them and their household members, and what they can do on their own to survive the various scenarios of a power failure crisis:

three days;
a week; and
over two weeks.

To this end, each household should stockpile an adequate level of drinking water, medicines, hygiene products and other necessities, as well as all of the equipment necessary to survive a prolonged power outage. Properly preparing a household for a prolonged blackout requires taking the following series of actions:

(1)    Identifying the number and categories of persons and animals living in the household and their daily needs;

(2) Defining the period of time for which stocks are to last. Preparing at once for 14 days of self-sufficiency will exceed the capacity of a novice. It is better to start by preparing well for three to five days; and

(3) Determining the range and minimum quantities of goods necessary for survival under conditions of prolonged power failure: water, food, medicines, hygiene, light, etc., first for one person, and then multiply this need by the number of people living in the shared household.

Self-responsibility requires that a minimum stock of drinking water, medicines, food and hygiene products, sufficient for at least two weeks, is maintained at all times, taking into account the needs of all household members. An adequate level of drinking water supplies is vital. The minimum daily requirement for one adult is 3 L. A supply equivalent to a "six-pack" of standard one-and-a-half-litre bottles should be collected for each person for three days. As for food, stocks are prepared comprised of products that do not require refrigeration (tinned food, pasta, rice). When buying, pay attention to the best-before date so as not to stock up on products with a short shelf life. In large cities, where the Western culture of eating out has become increasingly common, food stocks in modern households are very limited. In this respect, the situation is much better in small towns and rural areas where, due to lower incomes, larger living areas and, in many cases, the possession of home-grown food products, the custom of preparing all meals at home and stocking up on food for this purpose still persists. It is important to ensure that meals can be heated or cooked. A camping cooker (gas or other fuel) is a good option. It is recommended to have an adequate supply of fuel for such a cooker. You can also use the garden grill to heat up your meal. People on permanent medication or who are chronically ill should maintain a minimum stock of essential medicines sufficient for at least three weeks, as supply systems may need this much time to return to normal functioning after a blackout. It is also important to equip yourself with an emergency power system that will at least allow you to charge your phone or a rechargeable torch, and preferably a system with enough power and capacity to power the most important survival essentials: central heating, minimal lighting, a laptop, a radio, a small refrigerator and a coffee maker. There is a wide range of solutions on the market: from diesel generators to portable, suitcase-style Stromkoffer power systems, consisting of a battery, an AC inverter and a photovoltaic panel to charge the battery. Such suitcase-based emergency power supply systems provide up to 12 h of electricity to power the equipment you need most [59]. In sunny weather, the battery takes approximately 3 h to fully charge, and up to 6 h in cloudy weather. Such a system, including a photovoltaic panel, costs commercially around 800–900 Euros. To keep up to date with the latest developments and messages from the emergency services, it is also worth equipping yourself with a battery-operated radio. In most countries, there are statutory requirements requiring operators of critical infrastructures to maintain continuity of operation of energised systems for at least 72 h.

There will also be a demand for heat in winter. Even after a day without electricity, when the central heating is not working, the dwelling will get cold. This is when, not only warm clothing, but also a warm sleeping bag is needed. When the inside temperature approaches freezing, lighting a few candles in a small room will help to keep it above freezing point, which will also serve as a source of light and improve your mood, which is very important in a crisis situation.

Mutual neighbourly assistance is very important. A prolonged blackout will test people's responsibility for themselves and others, their social solidarity, self-help and mutual aid, and their capacity for civic self-organisation Self-help and mutual aid would be greatly facilitated by the rebuilding of community-based exchanges of mutual aid goods and services, enabling those offering help to contact those in need. In countries, such as Poland, an important role in the process of local community self-organisation and mutual assistance can be played not only by housing area authorities, but also by local associations and parishes.

*3.6. Insufficient Use of Social Capital Assets*

Without adequate preparation, emergencies resulting from a long-term blackout cannot be overcome on one's own, and in the case of a large-scale blackout, it is difficult to count on external assistance, because neighbouring administrations are usually preoccupied with dealing with the same problems and do not have the available capacity and resources. The basis for successful crisis management under conditions of a prolonged blackout is good preparation of everyone—both the administrative bodies and services responsible for ensuring public safety and the citizens themselves. In this context, it is particularly important not only to reliably inform citizens about the need to prepare for possible disruptions to supply systems and the need for self-help and mutual aid, but also to involve volunteers in specific tasks, e.g., organising and distributing humanitarian aid or protecting property, including strategically important facilities (e.g., fuel or material reserves, commercial outlets, etc.), from theft and other threats. However, in the documents reviewed, there was no stated intention to use the social assets of security—the knowledge, initiative, energy and self-organising capacity of citizens—in crisis operations. The refusal of crisis management bodies to make greater use of citizens' initiative, energy, knowledge, creativity and self-organisation capacities is incomprehensible in view of the critical inadequacy of resources—above all personnel reserves—which will be further significantly depleted by a prolonged blackout and the resulting paralysis of almost all areas of life. The involvement of volunteers and stakeholders is a source of mutual multidimensional benefits and should not be neglected in crisis situations—not only those of a complex and cascading nature.

Despite political declarations and programmes for the development of civil society, the idea of participation and opening the administration to the participation of citizens is still approached with great mistrust in Poland [60]. The reluctance of disaster management authorities to use citizens' initiative, to involve volunteers and to share tasks with local civil society organisations is due to a number of factors, most notably the fear of social conflicts that may arise in the face of shortages and a sense of insecurity. Observations made during emergencies challenge the widespread stereotype promoted by specialist state services according to which, in conditions of misfortune and scarcity, anarchy breaks out and rivalry between people intensifies. Contrary to the fears often expressed, in crisis situations the population behaves with more solidarity and cooperation and in a more disciplined manner than in everyday situations. At least in the initial phase of the crisis, the population shows attitudes of empathy, solidarity and readiness to help each other, cooperate and share. The tendency of people to act collectively and cooperatively in conditions of limited resources stems primarily from the awareness that, in various types of crisis situations, a community generally offers better opportunities for survival and a greater sense of security than acting alone on one's own. It is only with the passage of time, when dwindling resources threaten not to satisfy the most basic biological needs, that human conflicts arise. Thus, only an extreme shortage of basic products (e.g., water, food) can transform harmonious human interaction into violent conflict. The spectre of hunger may prompt people to break into disused commercial premises and loot them. In the event of extensive damage, the restoration of the supply function of such facilities will be delayed, which may subsequently jeopardise the security of a much larger proportion of the population. For this reason, the protection of such commercial premises against unauthorised entry by crowds is of great importance in crisis management, but local authorities rarely have the necessary human resources for this. The implementation of such tasks requires the involvement of the local community. Sometimes small policing teams formed from volunteers are sufficient for deterrence. In the chaos that accompanies emergencies, civilians want organisation and structure more than in everyday situations. Volunteers who enlist to help protect safety and order often only need to wear high-visibility waistcoats to increase the public's sense of safety and order.

## 4. Conclusions

On the basis of the research carried out, it is difficult to state categorically to what extent the Polish crisis management system is prepared to cope with the catastrophic effects of widespread and protracted power failures, because—apart from the recent COVID-19 crisis—such complex crisis situations have not yet occurred on the scale we are writing about. So far, power outages in Poland have been of a regional nature, and their cause has mostly related to physical damage to transmission infrastructures caused by adverse weather conditions. To date, the energy emergency services have been able to repair local or sectional damage and restore power supply to most of the affected areas within a few to a dozen or so hours, before a long-term, large-scale blackout could cause a humanitarian disaster resulting from the shutdown of critical infrastructure and the paralysis of social systems (security systems, transport systems, supply systems, etc.). However, crisis management should not be considered to be synonymous with emergency response, and especially not with the technical aspect of it, related to efficient damage control and rapid recovery [61]. In the framework of crisis management, public authorities seek to ensure the safety of the population living in their territory, using all available safety assets—not only the professional teams and specialised resources provided for in the crisis management plan, but also the knowledge, experience and wisdom of experts and the grassroots initiative, commitment, private resources and organisational capacity of citizens themselves. On the contrary, they contribute to an enhanced mobilisation, which may be decisive not only in the event of a catastrophic blackout, but also in the event of other multidimensional and large-scale emergencies, which the Polish population has not yet experienced.

With the dynamic increase in the complexity and unpredictability of the modern security environment, even with the best planning, it is impossible to build full preparedness for catastrophic events such as prolonged, large-scale power failures, which, under conditions of increasing population dependence on electricity-powered devices, can lead to the collapse of critical infrastructures and security systems, threatening humanitarian disasters and secondary threats that are difficult to control with limited crisis management resources. Effective preparation for crisis management in the conditions of a long-term, large-area blackout, which, fortunately, Polish authorities have not yet experienced, requires accurate, comprehensive, forward-looking assessment of the full spectrum of consequences of such events, building adequate scenarios for the development of crisis situations over time, taking into account different local conditions, as well as the identification of needs and challenges related to civil protection and the maintenance of operational continuity. In view of the vulnerability of most critical infrastructure systems to destabilisation in the event of long-lasting, large-scale blackouts and the grim spectre of spatio-temporally unlimited chains of harmful events threatening nation-wide catastrophes, as well as inadequate risk assessment of such crises and insufficient preparation of the Polish crisis management system for them, it is to be expected that the greatest challenge for crisis management in the event of such emergencies will be to overcome shock and maintain its own operational continuity.

To obtain a risk assessment for a long-term, large-area blackout that corresponds more closely to the actual level of the threat and to tailor more appropriate response scenarios in the process of crisis planning, one should strive to change the existing mechanistic, deterministic-linear paradigm dominant in the theory and practice of crisis management, based on the elementarisation of threats and calculability of risk, and to replace it with a systemic paradigm, a paradigm based on a comprehensive, organic, holistic view of an emergency situation that adequately takes into account its complexity and the dynamics of interactions, synergies and mutual reinforcement to which such situations owe their ability to spiral out of control and generate structurally and spatio-temporally unlimited chains of damaging events that spread at the speed of light and cannot be stopped by any barriers devised by man. However, under conditions of prolonged underinvestment in administrative safety and crisis management bodies and the lack of qualified staff—

particularly evident at the lowest levels of administration—these postulates appear to be extremely difficult to implement in practice.

Due to the rapid growth of the strategic importance of the energy sector, which is now becoming an increasingly important pillar of national security, an important domain of advantage-building and an arena of international conflicts, in the interests of internal security, development, integration and cooperation within the European Union, as well as in the interests of geopolitical strengthening position of the community, there is an urgent need to develop a common energy security policy in the EU based on the existing institutions. In view of the anticipated inefficiencies of national crisis management and civil protection systems in the face of surprising, cumulative crisis situations of mass scale and high escalating potential—such as blackouts—we recommend political decision-makers under the EU's common energy policy to initiate research and international and cross-sectoral consultations in search of solutions ensuring more effective protecting interconnected European power systems against blackouts and in the search for effective support mechanisms that will help member states build resilience and increase their ability to respond to such crises. In the face of the discussed challenges posed by the increasingly real threat of extensive and long-lasting blackouts to the security systems of the EU member states, the basis of resilience should also be effective mechanisms of international mutual assistance, without which it is difficult to rely on managing such complex, cumulative crisis situations only by using one's own buffers. Therefore, we propose the continuous improvement of the existing EU mutual assistance mechanisms under the EU crisis management system, taking into account the specific conditions of the Member States. The Emergency Response Coordination Centre (ERCC), handling the Civil Protection Mechanism, should play a special role [62]. The centre ensures the rapid deployment of emergency support and acts as a coordination hub between all EU Member States, the six additional Participating States, the affected country, and civil protection and humanitarian experts. The centre operates 24/7 and can help any country inside or outside the EU affected by a major disaster upon request from the national authorities or a UN body. We believe that the existing institutional solutions in the EU crisis management system are sufficient and there is no need to create additional bodies and structures, particularly when the EU's model dos not taking into consideration deeper federalization. All of the countries, as it was mentioned, are obliged to take responsibilities for national security, including people protection, according to the rules: "Domination of the territories system over central" and "Reaction on lower possible level". However, on the other hand it is impossible for any particular country alone to guarantee energy security and counter blackouts. This means that the EU should engage in planning activities and focus on risk assessments using own crisis management bodies that are equally responsible for external and inner security.

In the absence of experience with real catastrophic blackouts and in view of the limited possibilities to experiment with simulated blackouts, the basis for crisis scenarios and for building response scenarios should be analogue forecasting, based on our own experience with smaller-scale power failures (1), on our own experience with long-term, large-scale, complex cascading crises caused by other factors, such as the ongoing pandemic or the catastrophic floods of 1997 and 2010 (2), as well as on the past experience of other countries with catastrophic blackouts (3), taking into account the similarities and differences of national circumstances. In order for such an international exchange of experience and concepts for solutions to take place in a systematic way, it is worth considering the establishment of an international analytical and documentary research network specialising in interdisciplinary studies on blackouts, the improvement of crisis management concepts under conditions of long-term power shortages in large areas of Europe and the initiation of international cooperation for building population resilience to such crises. A European Research and Documentation Network on Blackout Resilience could gather information databases and exchange experiences with similar centres abroad. It would prepare annual reports containing information on incidents, their effects, response methods and assessment of their effectiveness; it would give opinions on the crisis management plans of public

authorities and enterprises, projects and undertakings from the point of view of blackout resilience; and it would conduct training courses for public authorities responsible for crisis management and for enterprises in the field of blackout preparedness.

With improved EU support mechanisms, the legal and institutional solutions and organizational structures used in the Polish crisis management system would create sufficient grounds for building social resilience to long-term, large-scale blackout and similar, cumulative, cascading crisis situations, provided that:

1. Risk assessment and planning will be improved, taking into account additional determinants, multivariate and flexible transitions between different modes of operation;
2. Networking and the use of systemic effects (self-organization potentials, synergy and feedback loops) will increase;
3. Intersectoral cooperation (e.g., between public administration authorities responsible for security, crisis management and civil protection and operators of critical infrastructures—primarily in the field of safety inspection, risk assessment and planning, determination of safety margins and buffers, information exchange, warning and alarming, handling dangerous incidents, assistance in maintaining business continuity and—if necessary—in reconstruction, crisis communication, humanitarian operations, etc.) will be strengthened;
4. The mobilization and involvement of additional social security assets (voluntary work, commercial security and safety services, etc.) will increase; and
5. Social campaigns to raise awareness of threats, promoting a culture of safety and responsibility for their own and common safety will be intensified, which should result in better preparation of the population to deal with long-term supply crises on their own (self-sufficiency, self-help, solidarity and sharing, etc.).

**Author Contributions:** Conceptualization: K.M., J.R.-Z. and D.M.; methodology: K.M.; programming: K.M.; validation: D.M. and J.R.-Z.; formal analysis: K.M.; investigation: K.M.; resources: K.M.; data handling: K.M.; writing—preparation of original draft: K.M.; writing—review and editing: D.M., K.M. and J.R.-Z. All authors have read and agreed to the published version of the manuscript.

**Funding:** This research received no external funding.

**Institutional Review Board Statement:** Not applicable.

**Informed Consent Statement:** Not applicable.

**Conflicts of Interest:** The authors declare no conflict of interest.

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
