# Peer review of "Readiness of the Polish Crisis Management System to Respond to Long-Term, Large-Scale Power Shortages and Failures (Blackouts)"

_energies, doi:10.3390/en14248286_

Round 1

Reviewer 1 Report

The paper analyzes a very important and timely problem and deserves to be published in Energies. Several changes can improve the presentation of the material: 

  • In the result session, before discussing planning errors, it would be be beneficial for reader to provide an overview of the Polish crisis management system.
  • The paper is very long so it is hard to follow the key issues. Some details can be moved to the supplement: about prolonged power system failures (lines 376-470, 620-635), Polish armed forces equipment (l 865-882) and disaster management plans.
  • The paper would benefit from streamlining the  policy recommendations for Poland and the EU. 

I am looking forward to reading the final version. 

Reviewer 2 Report

The work is based on a thorough background research. However, an overview of "good practice" in an international context is missing.

Reviewer 3 Report

The paper studies the solutions to improve the reliability of the power systems during power cuts and considers the readiness of the polish crisis management as a case study. The paper discusses interesting ideas and results. But some minor issues should be considered:

  • the use of quotation marks "" is extensive and inappropriate sometimes. (attacks) has been repeated in the abstract, line 24
  • please define CET in page 9, line 458
  • Page 10, the points in line 471, it is not clear where the bracket ends after: (e.g., dysfunctions in the energy transmission....
  • For all these points, they shouldn't be explained in the brackets.
  • Please check the previous points over the full manuscript.
  • Figures are required to describe the text and help the reader to follow up with some parts in the paper. For example, the systems described in the text of section 3.3.2 needs figures.
  • The final findings can be listed in the conclusion to help the reader.
